# Aspects of Phage-Based Vaccines for Protein and Epitope Immunization

**DOI:** 10.3390/vaccines11020436

**Published:** 2023-02-14

**Authors:** Marco Palma

**Affiliations:** 1Institute for Globally Distributed Open Research and Education (IGDORE), 03181 Torrevieja, Spain; marco.palma@igdore.org; 2Creative Biolabs Inc., Shirley, NY 11967, USA

**Keywords:** phage-based vaccine, phage vaccine, bacteriophage, vaccine, vaccination, immunization, protein-based vaccine, subunit-based vaccine, subunit vaccine, epitope vaccine

## Abstract

Because vaccine development is a difficult process, this study reviews aspects of phages as vaccine delivery vehicles through a literature search. The results demonstrated that because phages have adjuvant properties and are safe for humans and animals, they are an excellent vaccine tool for protein and epitope immunization. The phage genome can easily be manipulated to display antigens or create DNA vaccines. Additionally, they are easy to produce on a large scale, which lowers their manufacturing costs. They are stable under various conditions, which can facilitate their transport and storage. However, no medicine regulatory agency has yet authorized phage-based vaccines despite the considerable preclinical data confirming their benefits. The skeptical perspective of phages should be overcome because humans encounter bacteriophages in their environment all the time without suffering adverse effects. The lack of clinical trials, endotoxin contamination, phage composition, and long-term negative effects are some obstacles preventing the development of phage vaccines. However, their prospects should be promising because phages are safe in clinical trials; they have been authorized as a food additive to avoid food contamination and approved for emergency use in phage therapy against difficult-to-treat antibiotic-resistant bacteria. Therefore, this encourages the use of phages in vaccines.

## 1. Introduction

Bacteriophages or phages are viruses that replicate only within bacteria and are the predominant living forms on the planet [1]. They have various applications, e.g., in drug discovery [2,3], phage therapy [4], vaccine development [5], and biocontrol of pathogens in food products [6]. Phages can be a powerful tool for vaccine and antibody development since they are easy to produce and manipulate and can induce an immune response without adjuvants [5].

This review summarizes the main aspects of phage-based vaccines through a bibliographic search. It explains what a phage-based vaccine is and covers the various forms of phage-based vaccines and the most important phage types, and surface or capsid proteins for vaccine development. In addition, the steps of the production of phage-based vaccines are described. Important issues are explored, including whether phage-based vaccines are safe in humans and animals, and whether they work better than phage-free vaccines in eliciting an immune response. This study clarified whether there are ongoing or completed human clinical trials with phage-based vaccines and whether any phage-based vaccines have been authorized for use by medicine regulatory agencies. The advantages and disadvantages of phage-based vaccinations are discussed. Finally, their preclinical and clinical applications, the antigens or epitopes used in these studies, and whether these vaccines offer protection against the disease or condition are described.

## 2. Materials and Methods

### 2.1. Search Strategy

A literature review of articles addressing different aspects of phage-based vaccines was conducted by searching articles in PubMed (https://pubmed.ncbi.nlm.nih.gov/ (accessed on 5 January 2023)) using the keywords phage OR bacteriophage OR phage-based OR bacteriophage-based AND vaccine in the title/abstract. Articles published from 1 January 2017, to 15 December 2022, were analyzed.

### 2.2. Study Selection, Inclusion, and Exclusion Criteria

Studies that were not conducted in English were also discarded. Letters, abstracts from conferences, and editorials were also not included. Studies without an animal model or clinical trials were also omitted, as were studies that did not use phages for immunization. Additionally, articles clarifying each disease or condition, antigen, and epitope were included.

## 3. Results

### 3.1. Description of Phage-Based Vaccines

Phage-based vaccines use phages as carriers to present antigens to the immune system to generate a proper immune response for protection against a specific disease [7]. Currently, there are three types of vaccines based on bacteriophages: display, DNA, and hybrid vaccines [5,8]. In phage-displayed vaccines, the phages express proteins or protein subunits such as peptides with one coat protein or the antigens are chemically conjugate to the phage surfaces. Moreover, peptide screening using monoclonal or polyclonal antibodies from patients with a disease can reveal specific regions that can induce a neutralizing response [9,10]. In phage-based DNA vaccines, a gene fragment encoding a protein or a protein subunit is cloned into a eukaryotic cassette in a plasmid and then packed into phage particles. The phage particles are uptaken by target cells that express the antigen, leading to an immune response. The hybrid phage vaccine is a fusion of two types of phage vaccines.

For the production of vaccines, several lytic and filamentous phages have been used, especially those that replicate in *E. coli* [11]. In theory, any bacteriophage is suitable for phage-based vaccines, but they should meet some criteria to facilitate phage construction and production. The sequence of the phage genome and the genes encoding the capsid and tail proteins should be known and can be modified to promote the display of foreign antigens [12]. The incorporation of the target sequence should not affect the phage stability and assembly. The phage bacterial host should be a nonpathogenic laboratory strain that can be used on a standard laboratory bench. Moreover, an alternative approach could be used, which applies to any bacteriophage, and antigens can be covalently linked to the bacteriophage surface by a chemical method such as the SpyCatcher system. However, the toxicity of untested bacteriophages should be analyzed first. One of the main reasons that *E. coli* phages are used for phage-based vaccines is that they are well-known, easy to use, and have already been tested in animal models and clinical trials [13]. In addition, many well-developed tools and protocols are available for engineering these phages. The most common phages used in the selected studies were lambda, Qβ, T4, MS2, T7, AP205, M13, and M13KE, which are listed in Table 1.

Several proteins, including E, D, B, W, FII, B, X1, and X2, make up the phage lambda’s head, whereas at least six proteins, including U, V, J, H, Tfa, and Stf, make up the tail [14]. Both the head protein pD and the tail protein pV can expose antigens on lambda [15,16]. Two capsid proteins (gp10A and gp10B) make up the icosahedral head of T7. gp10B has been used in the first place since it is not required for the assembly of the capsid [17,18,19].

The bacteriophage T4 capsid is composed of gp23, gp24, and gp20, which are put together by the two capsid proteins (Soc and Hoc) [20]. Given that Soc and Hoc are not necessary for phage infectivity, they have been used to express antigens on the surface of T4 [21,22].

Phage M13 has a major coat protein (pVIII) and several minor capsid proteins (pIII, pVI, pVII, and pIX). At one end of the phage are the proteins pIII and pVI, while at the other are pVII and pIX. M13 specifically infects *E. coli* that have F pilus [23]. pVIII has been used to display peptides, while pIII to display larger fragments (scFv and Fab) in addition to peptides (Figure 1). Due to the high copy number of pVIII on the phage surface (2700), pVIII is preferred to display multiple copies of an epitope in a phage-based vaccine. F8 is a filamentous bacteriophage similar to M13 regarding both its genome and proteome [24,25].

The bacteriophages Qβ, MS2, and AP205 have been used to create virus-like particles (VLP). VLPs are assembled viral proteins at the nanoscale that are not contagious because they lack viral genetic material [26]. Chemical conjugation and genetic fusion are the two main methods to display immunogenic antigens on VLPs [27]. Qβ encodes three main proteins (CP, A2, and a replicase), [28] while MS2 has CP, A, L, and a replicase [29]. In these two bacteriophages, antigens have been fused to their surface coat proteins to develop vaccines against HIV [30] and drug addiction [31,32]. AP205 is a single-stranded RNA (ssRNA) bacteriophage specific for Acinetobacter bacteria. This bacteriophage has 180 copies of the coat protein that allows both N- or C-terminal fusion of epitopes up to at least fifty-five amino acids [33].

**Table 1 vaccines-11-00436-t001:** Commonly used phages in phage-based vaccines.

Phage	Characteristic	Application	Ref.
Lambda	Icosahedral, dsDNA, *E.coli*	Display	[15,34,35]
T7	Icosahedral, dsDNA, *E.coli*	Display	[36,37]
T4	Icosahedral, dsDNA, *E.coli*	Display	[38,39,40]
M13, M13KE	Filamentous, ssDNA, *E. coli*	Display	[41,42,43]
F8	Filamentous, ssDNA, *E. coli*	Display	[44,45]
Qβ	Icosahedral, ssRNA, *E.coli*	Conjugated	[46,47]
MS2	Icosahedral, ssRNA, *E.coli*	Conjugated	[48,49]
AP205	Icosahedral, ssRNA, *Acinetobacter*	Conjugated	[50,51]

Phage-based vaccine preparation entails several steps, including the creation of plasmids, the transformation of *E. coli*, phage propagation, removal of bacterial cells, phage precipitation and centrifugation, pellet resuspension, endotoxin removal, filtration, dialysis, and aliquotation of the vaccine stock (shown in Figure 2).

The propagation process for each phage and bacterial host may be different than another. Here is a brief description of the production of the most commonly used phage in phage-based vaccine development, the lysogenic filamentous phage M13, which can infect trains of F+ male *E. coli* through their pili (e.g., TG- 1). To display the target antigen on the surface of filamentous phages, a phage vector (e.g., fUSE5, fAFF1, fd-CAT1 or fd-tet-DOG) or a phagemid [52] (e.g., pHEN1, pComb3, pComb8, or pSEX) is used. A phage vector is composed of almost complete phage genome (e.g., M13), into which DNA encoding the desired antigen is introduced in the gene encoding a capsid protein (e.g., pVIII or pIII). The remaining portion of the phage genome typically remains untouched and generates the additional gene products required for the phage life cycle. The phage vectors allow multivalent expression of the target antigen, also displayed in fusion with every copy of the chosen capsid protein. A phagemid, on the other hand, is a plasmid that carries only the gene for one of the phage capsid proteins. Therefore, a helper phage (e.g., M13KO7 and VCSM13) must provide the remaining phage proteins for phage production.

After being constructed, the phage vector or the phagemid is introduced into TG-1 by electroporation or chemical transformation.

For phage propagation from bacterial cells with phagemid, an overnight culture is used to inoculate a sterile growth medium in an Erlenmyer flask and grow at 37 °C to mid-log phase, at which point helper phage is used to infect the bacterial cells in the ratio of 1:20 and incubate in falcon tube without shaking in 37 °C water bath for 30 min. The infected cells are collected by centrifugation. Then, the supernatant is removed, with the pellet resuspended in a fresh medium containing antibiotic and incubated at 30 °C overnight on a shaker in a baffled flask. For propagating phages from bacterial cells with phage vectors, the step with helper phages is omitted.

Bacterial cells are removed from the culture containing the recombinant phages by centrifugation twice and discarding the pellets. Then, the phages are collected by treatment of the supernatant with one fifth the volume of 20% PEG/2.5 M NaCl for 2 h at 4 °C and centrifugation. The supernatant is then discarded, the pellet is reconstituted in sterile phosphate-buffered saline (PBS), and the PEG/NaCl procedure is carried out once more.

To employ phage preparations in vaccines, they must be cleaned of bacterial endotoxins, which are frequently present when it uses bacterial cultures. According to Bordier (1980), contaminating endotoxins are eliminated by repeated two-phase Triton X-114 separation [53]. The cleaned bacteriophages are dialyzed against sterile PBS after passing through a 0.4 m nitrocellulose filter. Other endotoxins removal methods could be used such as the one with organic solvents described by Szermer-Olearnik and Boratyński (2015) [54].

The quality of the stock needs to be checked before using it by determining its titer and endotoxin levels. Then, it can be diluted to the working concentration.

The prepared phage stocks can then be aliquoted into cryotubes as working stocks and temporarily stored at 4 °C. For long-term storage, glycerole is added to a final concentration of 20%, which can be kept at −80 °C.

Before a frozen phage stock is used, the glycerol needs to be removed by treating it with PEG/NaCl and centrifugation, and then resuspending the pellet in sterile PBS.

### 3.2. Safety of Phage-Based Vaccines for Humans and Animals

Human studies have shown that phage-based vaccinations have no notable negative effects on humans. Phage particles cannot replicate in eukaryotic cells. Therefore, they are significantly safer than other viral vaccines [10]. Bruttin and Brüssow did not record adverse outcomes in healthy adult volunteers that received low doses of *E. coli* phage T4 in their drinking water [55]. A study by Algazi and colleagues evaluated the safety and efficacy of the phage vaccine SNS-301 along with pembrolizumab in people affected by chronic myelomonocytic leukemia (CMML) and high-risk myelodysplastic syndromes (MDS). The results indicated that all doses of PAN-301-1 were safe, and no drug-related adverse events or dose-limiting toxicities occurred during the study [15]. In another trial, fifteen patients with advanced multiple myeloma well-tolerated anticancer vaccine based on phage M13K07 [13]. Sarkere et al. reported no adverse effects in healthy and diarrheal children from Bangladesh who received oral administration of T4-like coliphage cocktail preparations [56,57]. The phage study by Febvre and colleagues showed that oral administration of coliphages does not affect the fecal microbiota composition in healthy adults [58].

Preclinical studies have demonstrated that phages are safe for animals. Johnson et al. assessed the safety of phage-based vaccines for fertility control in cats. They did not show any signs of local or systemic reactions [45]. In another study, Qβ bacteriophage virus-like particles conjugated with a peptide corresponding to a region of the human L-type calcium channel (epitope CE12) had no adverse effects on immunized rats [46].

The safety results of phages in phage therapy could indicate similar safety for phage-based vaccines, using the same type of phage under similar conditions and similar phage preparation. An extensive review of phage safety and toxicity could be found in the work of Liu and colleagues [59].

### 3.3. Elicitation of the Immune Response by Phage-Based Vaccines

Bacteriophages stimulate both innate and adaptive immunity [12]. Antigen-presenting cells express toll-like receptors (TLR) that identify non-self antigens, such as phages at the early stages of the immune response [60], which are internalized by these cells for maturation and presentation. Moreover, the peptides bind to the major histocompatibility complex (MHC) class I or II, which are then identified by CD8+ or CD4+ T-cells, respectively [61]. CD4+ T-cells promote antibody expression by B-cells (Th2 cells) and cytotoxicity by CD8+ T-cells (Th1 cells) [62].

Phages-based vaccines commonly do not require adjuvants in their composition because bacteriophages are effective adjuvants, capable of increasing the immunological response to any antigen administered together with the phage particles. Unmethylated deoxycytidylate-phosphate-deoxyguanylate (CpG) dinucleotide in the phage genome induces this adjuvant effect [63]. Further, the unmethylated CpG nucleotides can induce murine B-cells to proliferate and secrete immunoglobulin [64], induce cytokines, stimulate natural killer cells, and elicit T-cell responses [65].

Recently, Davenport and colleagues demonstrated that the recombinant receptor-binding domain (RBD) of the severe acute respiratory syndrome (SARS) alone was insufficient to induce RBD-specific IgG in mice. However, RBD_SARS_- phage-like particles from lambda decorated with RBD not only elicited high levels of RBD_SARS_-specific IgG, but they remained high for 174 days [34]. In the study by Ortega-Rivera et al., mice immunized with Qβ conjugated with the S protein peptides TESNKKFLPFQQFGRDIA, PSKPSKRSFIEDLLFNKV, HADQLTPTWRVY produced high-level IgG antibodies, while animals vaccinated with free S protein peptides did not [47]. Li and colleagues found that whereas the proteins of influenza virus 3M2e displayed on T4 nanoparticles, without any adjuvant, elicited extremely high-levels of 3M2e-specific IgG antibodies, 3M2e conjugated to RB69 Soc induced low levels of 3M2e-specific IgG antibodies [38]. In addition, the 58 amino acid peptide of the Pfs47 antigen of *P. falciparum* conjugated phage AP205 VLP was more immunogenic than unconjugated Pfs47 monomers, inducing a high level of antibodies [51]. VLPs of bacteriophage Qβ displaying the target peptides PCKS9, ApoB, and CETP induced higher antibody titers in mice against the antigen than the free peptides [66].

In some studies, an adjuvant was also used along with phages [42,67], perhaps because they needed a higher immune response than that obtained with only the phages.

### 3.4. Phage-Base Vaccines in Clinical Trials

A clinical phase I/II trial (NCT04839146) evaluated in healthy volunteers the safety of ABNCoV2, a vaccine based on VLPs of bacteriophage AP205 decorated with RBD of SARS-CoV-2 produced in S2 Drosophila cells [50]. This study ended on 25 February 2022, but no results have been published at clinical trials.gov. In addition, an open-label phase two trial (NCT05077267; EUCTR2021-001393-31) is ongoing with ABNCoV2 in Germany. A phase three trial (NCT05329220) evaluated the immunogenicity, safety, and tolerability of ABNCoV2 in adults previously vaccinated with SARS-CoV-2.

Human aspartate β-hydroxylase (ASPH) is associated with various types of cancer, and it has been well-tolerated in vaccines, inducing an immune response. SNS-301 is a bacteriophage lambda that displays a part of ASPH fused inframe with the coat protein gpD. A clinical trial assessed the safety, immunogenicity, and preliminary clinical efficacy of the phage-based vaccine SNS-301 (clinicalTrials.gov, identifier NCT04217720, and NCT04034225). People affected by chronic myelomonocytic leukemia and high-risk myelodysplastic syndromes received SNS-301 together with pembrolizumab. The results showed that the combination of SNS-301 with pembrolizumab was well-tolerated among patients and resulted in disease stabilization and tumor response. Because these are preliminary results, further studies on the efficacy of the combination therapy are needed [15].

### 3.5. Approved Phage-Based Vaccines

The United States Food and Drug Administration (FDA) and The European Medicines Agency (EMA) have not yet approved any phage-based vaccinations, regardless of the promising results found in preclinical models and the fact that phage-based vaccines are safe for humans and animals. However, the FDA has given the go-ahead for bacteriophages to be used as antibacterial agents against contaminated food, e.g., against Listeria monocytogenes in ready-to-eat meat and poultry products. Because phages only infect bacteria and not mammalian or plant cells, the FDA concludes that the food additive under review does not pose a toxicological risk for use in food. Phages are also part of the normal microbial population of the human gut, and humans are often exposed to them in high amounts through food, water, and the environment without suffering any negative effects [68].

On 19 February 2019, the FDA authorized a U.S. clinical trial of intravenous bacteriophage therapy [69]. In addition, phagebank therapy in COVID-19 patients with suspected or proven secondary bacterial infections received FDA approval under the Emergency Use Authorization (EUA) [70]. Several case reports have described the use of phages to treat persistent bacterial infections after multiple failures of antibiotic treatments [71,72,73,74,75,76,77,78]. These studies are encouraging since this could indicate that bacteriophages may be safe in other medical applications, including in phage-based vaccines, without being a risk to people or animals [79].

### 3.6. Advantages and Disadvantages of Phage-Based Vaccines

Phage-based vaccines have many advantages compared with only a few disadvantages and are generally well-suited as vaccine tools with many properties. Phages are safe for both humans and animals. They can be easily genetically modified [80] and trigger a strong humoral and cellular immune response. Therefore, adjuvants are not required [81]. They do not need to be produced in cell culture systems and can be easily converted for mass production using basic bacteriological media [82]. Phage-based vaccines could be a simple, time- and cost-efficient vaccination approach [13]. Bacteriophages may also be applied to oral vaccination due to their physical stability in the gastrointestinal tract [83]. Phages have a significant potential for use as vaccine carriers for various illnesses, including cancer and infectious diseases, which are listed in Table 2 [12,84]. Viral particles are remarkably stable in many challenging environmental conditions (pH and temperature) [10]. These characteristics make administration, storage, and transportation easy [8]. For instance, some stability tests revealed that T4 was stable for at least ten weeks at room temperature. Therefore, a cold chain is not required to distribute vaccines based on T4 [40]. Phage DNA vaccines allow gene expression and protein folding in eukaryotic cells, and the eukaryotic expression cassette is protected from degradation [85].

Phage-based vaccines have some potential limitations that need to be mentioned and considered when developing them. For example, it can be challenging to correctly display a molecule on the phage surface, resulting in missing active epitopes to elicit a meaningful immune response [84]. In addition, the antigen size can be a limitation because it is unlikely that large protein subunits could be displayed on phage particles. However, this is solved by conjugating the phages with these large antigens. Additionally, for the phage DNA vaccines, the genome length must be within the virion packaging limits [86]. The lack of an immunological response to self or harmful antigens may also be a barrier [87]. Another disadvantage is that few clinical trials support the efficacy of phage-based vaccines, and more efforts must convert the results from preclinical investigations to clinical trials.

Endotoxins (lipid A), released during the replication of lytic phages [59] or derived from the host bacteria and culture media, can contaminate the production of phage-based vaccines. Endotoxins are also the hydrophobic moiety of lipopolysaccharide (LPS), which makes up the outer monolayer of the outer membranes of most gram-negative bacteria [88]. Endotoxins can be an issue since even slight exposure can trigger pro-inflammatory reactions [88] and lead to toxic shock, cell damage, and the production of cytokines [89]. Due to these effects, the generated phage stock must undergo a procedure to remove endotoxins before they can be used in vaccines. This can be minimized using lysogenic phages, such as filamentous phages (e.g., M13, fd, and f1), instead of lytic phages and implementing good manufacturing practice (GMP) [90] in an early stage of the product development to meet regulatory requirements [91]. The removal of endotoxins from bacterial cultures can be done using many techniques, such as triton X-114 separation [53] and organic solvents [54]. The triton X-114 separation worked quickly and effectively to prepare M13 phage-based vaccine by lowering the endotoxin concentration to less than one unit/mL, which was used in patients with advanced multiple myeloma participating in a clinical phase I/II trial. The triton X-114 treated phage vaccination was well-tolerated in the patients, with the only minor side effects being flu-like symptoms and skin irritation at the injection site [13].

**Table 2 vaccines-11-00436-t002:** Applications of phage-based vaccines.

Condition	Agent	Target Antigen	Phage	Ref.
Viral infections	SARS-CoV-2	S protein, NP	Lambda, AP205, QβBxb1, Bxb2	[34,39,40,47,67,86,92]
	Influenza A virus	3M2e	T4	[38]
	Dengue virus	NS1	Qβ	[93]
	Foot-and-mouth disease	VP1	MS2, T7	[36,48]
	Zika virus	E protein	Qβ	[36]
Bacterial infections	Chlamydia trachomatis	CT584, MOMP	Qβ, MS2	[49,94]
	Escherichia coli	STh	AP205	[95]
	Vibrio Cholera	OSP	Qβ	[96]
	Bordetella pertussis	Pentasaccharide	Qβ	[97]
Parasite infections	Rhipicephalus microplus	Sbm746	M13	[41]
	Fasciola hepatic	Cathepsin	M13KE	[42,98]
	Plasmodium falciparum	Pfs47	AP205	[51]
Yeast infections	Candida albicans	Fba1	M13	[99]
Cancer	Breast cancer	HER2, xCT	Lambda, M13, MS2	[35,43,100,101]
	Leukemia	ASPH	SNS-301	[15]
	General	GD2	Qβ	[102]
	Metastasis and solid tumor	MUC1	Qβ	[103]
	Melanoma	Neoantigens	T7	[37]
Other conditions	Tauopathies	Tau	Qβ	[104]
	Hypertension	CaV1.2	Qβ	[46]
	Allergic rhinitis	H4R	M13	[105]
	Cardiovascular diseases	ApoB, PCSK9, and CETP	Qβ	[66]
	Fertility control	Gonadotropins	f8	[45]

### 3.7. Phage-Based Vaccines against Viral Infections

#### 3.7.1. SARS-CoV-2

The SARS coronavirus-2 (CoV-2) pandemic began to expand worldwide in 2019, killing more than 6600 million people worldwide by December 2022 [106]. The RBD plays a role in the binding of SARS-CoV-2 to ACE-2, allowing the virus to enter the host cells [107]. Moreover, antibodies can neutralize this attachment by competing with ACE-2 for binding to RBD. Therefore, the spike (S) protein is the main target to develop a phage-based vaccine.

Phage-based vaccine named RBD_SARS_-PLPs or RBD_SARS_-PLPs were developed by decorating phage-like particles (PLPs) of phage lambda with recombinant RBD from SARS-CoV-2 or MERS-CoV, respectively. PLPs decorated with a combination of RBD from SARS-CoV-2 and MERS-CoV were called hCoV-RBDs-PLP. Recombinant RBD proteins containing the residues 319–541 of the S protein from the SARS-CoV-2 Wuhan-Hu-1 or residues 367–588 from the MERS-CoV EMC/2012 were crosslinking to the lambda mutant protein gpD (S42C). Subsequently, the gpD-RBD constructions conjugated to purified lambda PLPs. RBD_SARS_-PLPs triggered a strong antibody-mediated response specific for RBD in BALB/c mice and protected them from the SARS-CoV-2 challenge. BALB/c mice vaccinated with hCoV-RBD PLPs produced serum-neutralizing antibodies that protect them against SARS-CoV-2 and MERS-CoV lung infections [34].

The S protein S1 domain (residues sixteen to 679) was chemically conjugated to VLPs of bacteriophage AP205 using SpyTag/SpyCatcher system. The SpyTag/SpyCatcher system forms an irreversible covalent bond between a peptide (SpyTag) and a protein partner within minutes [108,109]. Immunized mice produced higher levels of specific IgG after the first injection with S1- VLP than mice injected with S1 alone. However, the levels of these antibodies were almost equal after the second injection due to the presence of AddaVax, an MF59-like oil-in-water emulsion. Their sera could inhibit two SARS-CoV-2 variants (e.g., Wuhan and UK/B.1.1.7) in vitro [67].

Three linear, conserved neutralizing epitopes from the S protein, namely S14P5 or 570 (TESNKKFLPFQQFGRDIA), S21P2 or 826 (PSKPSKRSFIEDLLFNKV), and S1-105 or 636 (HADQLTPTWRVY), were isolated using sera from convalescent patients. These were then conjugated to the bacteriophage Qβ to produce Q570, Q636, and Q826. A single dose of the phage vaccine in a slow-release polymer implant injected in mice induced a strong antibody response to the three epitopes and S protein, which could neutralize the attachment of SARS-CoV-2 to human cells [47].

Another vaccine approach uses two different peptide antigens on the surface of bacteriophages. The S protein epitope 4 (aa 662–671) of SARS-CoV-2 was displayed with the coat protein pVIII of phage f88–4, while the peptide CAKSMGDIVC was with the capsid protein pIII. CAKSMGDIVC is a lung transport peptide that can recognize lung epithelium cells [110]. Mice immunized with phages displaying S protein epitope generated a robust antibody response to the S protein. This study demonstrated that the pulmonary delivery of vaccines using a lung transport peptide can trigger both local and systemic immune responses [92].

Fragments of the S protein were displayed on mycobacteriophages Bxb1 to create a phage-based vaccine. Two bacteriophage Bxb1-based vaccines were developed using the CRISPY-BRED technique: one that displays SARS-CoV-2 antigens (BaDAS); and another vaccine that DNA encodes and displays antigens (DEaDAS). The two vaccine candidates stimulated an immune response against the S protein in mice, but the generated antibodies could not neutralize the SARS-CoV-2 virus in vitro [86].

In addition, vaccines containing other SARS-CoV-2 antigens than the S protein have been designed. The ectodomain trimers of the S protein were expressed on the T4 phage, forming T4-CoV-2, while the nucleocapsid protein (NP) was in the interior of the phage. Intranasal injection of a vaccine against SARS-CoV-2 increased the immune response on the mucosal surfaces compared. However, this was not observed with the intramuscular injection. This was also observed in human angiotensin-converting enzyme (hACE2)-transgenic mice. These mice became protected against a deadly dose of the original SARS-CoV-2 and its Delta variant. Additionally, this vaccine was stable at room temperature and did not affect the microbiome [40].

Zhu et al. developed a platform using CRISPR editing of phage T4 to generate several candidates for a SARS-CoV-2 vaccine. Different SARS-CoV-2 antigens (e.g., S, E, and NP proteins) were integrated into separate compartments of the phage. T4 decorated with S-trimers evoked specific antibodies and blocked RBD binding to the ACE-2, protecting mice from SARS-CoV-2 infection [39].

#### 3.7.2. Influenza A Virus

Influenza A (Flu) virus is a very contagious virus that can cause dangerous respiratory infections [111]. New vaccines against the Flu were developed using a platform based on T4 phages. Peptides consisting of twenty-three amino acids from the amino terminal part of the extracellular domain of influenza matrix protein 2 (3M2e) from human, swine, and avian influenza viruses were displayed on phage outer capsid protein (Soc), forming the 3M2e-T4 VLP vaccine [38]. This vaccine given intramuscularly to mice without any adjuvant triggered strong cell- and antibody-mediated immune responses. The produced antibodies were specific and highly immunogenic against M2e on influenza virions and virus-infected cells, which protected mice challenged with the H1N1 virus. These experiments demonstrate the feasibility of the phage T4 VLP platform to develop the next-generation influenza vaccines [38].

#### 3.7.3. Dengue Virus

A bacteriophage VLP platform was developed to design a Dengue virus (DENV) vaccine. Different synthetic peptides corresponding to conserved areas from four serotypes (DENV1-4) of DENV’s non-structural protein 1 (NS1) were chemically attached to VLPs derived from bacteriophage Qβ. BALB/c mice were vaccinated intramuscularly at three-week intervals with the Qβ-NS1-PEPs vaccine with no exogenous adjuvant. Vaccinated animals generated antibodies that identified highly conserved epitopes of NS1 from all four DENV serotypes and DENV-2-infected cells. These findings suggest that an epitope-specific vaccination targeting conserved NS1 areas may be a potential method for DENV vaccines or treatments aimed at binding circulating NS1 protein [93].

#### 3.7.4. Foot-and-Mouth Disease Virus

Foot-and-mouth disease (FMD) affects cloven-hoofed animals, resulting in massive economic losses worldwide [112]. FMD virus (FMDV) has four proteins in its capsid (VP1-4), of which one of them (VP1) has an RGD sequence in its G-H loop that mediates attachment to the integrin. This loop has a major antigenic epitope capable of eliciting FMDV-specific neutralizing antibodies [113]. Phage-based vaccine against FMDV was constructed by displaying a fragment corresponding to residues 131–160 of VP1 on the surface of the MS2 phage, which was termed the chimeric nanoparticle (CNPs) vaccine. Mice immunized with CNPs generated higher specific antibodies against VP1 and increased the cellular immune response much more than the control peptides. These findings showed that the CNPs developed for the current investigation might serve as a viable replacement vaccination for FMDV control in the future [48]. In another study, bacteriophage T7 was used to design vaccines to prevent infections from several types of viruses, including influenza A and FMDV. VP1 was displayed on phage T7 generating the vaccine AKT-T7, which was not toxic to sheep kidney-, BHK-21-, or MDBK-cells. The immunized mice developed high levels of FMDV antibodies that could be sustained over time. Furthermore, the animals produced significant amounts of IFN-γ in mice while having minimal effect on IL-4 [36].

#### 3.7.5. Zika Virus

Unborn children of pregnant women may be affected by microcephaly or other congenital malformations if their mothers are exposed to the Zika virus (ZIKV) during their pregnancy. The E protein is the major surface glycoprotein of ZIKV, which is responsible for virus attachment and fusion into the host cells. Because this protein is the main target for generating neutralizing antibodies [114], Phage-based vaccine was developed by expressing B-cell epitopes of the ZIKV E protein on VLPs of bacteriophage Qβ. Immunized mice with phages with a single epitope generated antibodies against ZIKV. However, these antibodies only protect against a small viral inoculum. In contrast, a cocktail of phages displaying different B-cell epitopes of ZIKV E not only triggered a strong antibody response but also neutralized the viral infection at higher doses. As a result, vaccination with various VLPs exhibiting numerous ZIKV B-cell epitopes is an effective technique for increasing ZIKV neutralization [115].

### 3.8. Phage-Based Vaccines against Bacterial Infections

#### 3.8.1. *Chlamydia trachomatis*

*Chlamydia trachomatis* can be found in the mouth, genital tract, cervix, and urethra of adults. This bacterium is a leading cause of sexually transmitted infection, affecting more than 120 million people worldwide [116]. CT584 is a protein linked with the type three secretion system (T3SS) that plays a role in invasion, intracellular survival, and exit from cells [117]. CT584 forms part of a vaccine composed of three proteins that induce the production of neutralizing antibodies that protect against *C. trachomatis* [118]. Moreover, the immunogenicity and effectiveness of the phage-base vaccine in preventing *C. trachomatis* infections were evaluated. Epitopes 70–77 and 154–164 of CT584 were chemically conjugated to VLPs of bacteriophage Qβ, forming the Qβ-CT584 vaccine. Mice immunized intramuscularly with Qβ-CT584 and then challenged with *C. trachomatis*, produced high levels of IgG against CT584 that protected them from infection. CT584 antibodies were also found at the site of mouse infection. Moreover, the pre-incubation of *C. trachomatis* with sera from mice immunized with Qβ-CT584 reduced the number of bacterial cells in the upper genital tract of challenged animals. Surprisingly, sera from women affected by urogenital *C. trachomatis* infection did not identify the CT584 epitopes employed in the vaccine. However, species-specific differences in immunogenicity between humans and mice require further investigation [94]. A transmembrane protein called MOMP, for major outer membrane protein, has four variable domains exposed to the surface and five constant domains (VDs). MOMP has a highly conserved epitope (TTLNPTIAG) in the domain 4 (VD4) [119] that elicits neutralizing monoclonal antibodies (E4 mAb) to *C. trachomatis* [120]. The phage-based vaccine was developed by selecting phages from a peptide library with E4 mAb. An alternative was to express the TTLNPTIAG epitope on the surface of the VLPs of bacteriophage MS2. In a mouse challenge model, mice immunized with MS2 VLPs generated IgG that identified *C. trachomatis* and gave protection against vaginal Chlamydia infection [49].

#### 3.8.2. *Escherichia coli*

The native human heat-stable toxin (STh) and its variant STh-A14T toxoid are important virulence factors of enterotoxigenic *E. coli* (ETEC). *E. coli* phage-based vaccine was prepared by coupling STh and STh-A14T to the surface of bacteriophage AP205 by applying the SpyCatcher system. Mice were injected intramuscularly with the AP205 VLP vaccine without adjuvants to induce antibody production that could block the toxic activity of native STh. In addition, these antibodies recognize STh, without cross-reacting with endogenous guanylyl cyclase C (GC-C) receptor peptides uroguanylin and guanylin, which have sequence and structural similarity to STh. Additionally, the mutation in the STh-A14T variants did not affect the immunogenicity of the neutralizing ability of the resulting sera compared with native STh. Together, their results show that VLPs are effective STh immunogen delivery vehicles and that the STh-A14T-conjugated AP205 VLP is a potential ETEC vaccination candidate [95].

#### 3.8.3. *Vibrio cholera*

*V. cholera* is a very contagious bacteria that causes an acute, secretory diarrheal disease known as cholera. Its serotype classification is based on the O antigen of the lipopolysaccharide (LPS). Of more than 200 serogroups, only serogroups O1 and O139 may cause epidemic cholera [121]. A vaccine (Qβ-OSP) against *V. cholera* was designed by chemically connecting the antigen O-specific polysaccharide (OSP), recovered from *V. cholerae* O1 El Tor Inaba to VLP derived from phage Qβ. Immunizing mice with the Qβ-OSP formula induced high and long-lasting levels of IgG antibodies against OSP, and the resultant antibodies recognized the natural LPS from *V. cholerae* O1 El Tor Inaba. Additionally, antibodies from sera from immunized mice with complements could lyse the live bacteria [96].

#### 3.8.4. *Bordetella pertussis*

*Bordetella pertussis* causes whooping cough or pertussis, a severe paroxysmal coughing condition, mainly in infants and children [122]. Lipopolysaccharides (LPS) are part of the outer membrane of Gram-negative bacteria, exhibiting endotoxic activities [123] that make them attractive antigens to develop vaccines against *B. pertussis*. The LPS of *B. pertussis* includes lipid A coupled to dodeca-saccharide [124] that are recognized by monoclonal antibodies, suggesting that it may be a suitable epitope target in vaccines [125]. Phage-based vaccine against *B. pertussis* was developed by conjugating this pentasaccharide to the bacteriophage Qβ (Qβ-glycan). Immunization of mice with Qβ-glycan induced robust and long-lasting anti-glycan IgG that could recognize native LPS on the bacterial surface [97].

### 3.9. Phage-Based Vaccines against Parasite Infections

#### 3.9.1. *Rhipicephalus microplus*

The immune response of a candidate phage-based vaccine against *R. microplus* (cattle tick) was evaluated. The vaccine was constructed by displaying the epitopes Sbm7462 from *R. microplus* Bm86 protein on M13 phage, fused to a truncated pIII protein. Sbm7462 phage vaccine triggered the maturation of bovine monocyte-derived dendritic cells in an ex vivo assay. Subcutaneously vaccinated mice showed proliferation of peripheral blood mononuclear cells (PBMC) from the spleen and production of antibodies against Bm86 and Sbm7462 antigens. The suggested vaccination generated a similar immunological response as the currently available Bm86-based vaccine. Although the ex vivo assay provides a broad picture of the vaccine’s effects, a bovine in vivo experiment is still required to establish the vaccine’s efficacy [41].

#### 3.9.2. *Fasciola hepatic*

One of the main pathogens of fasciolosis in cattle and sheep is *F. hepatica*, which causes enormous economic losses to the livestock industry. No suitable commercial vaccines are available because of the immune evasion strategies of these parasites [126]. Cathepsin L1 and L2 (CL1 and CL2) are cysteine proteases that induce parasite penetration into host tissues. Therefore, they might be a good vaccine target against this parasite. Different mimotopes of CL1 and CL2 displayed on phage M13KE were used as vaccine candidates against *F. hepatic*. Goats vaccinated with phages with the mimotopes DPWWLKQ and SGTFLFS of CL1, and PPIRNGK of CL2 with Quil A adjuvant generated high levels of Ig1 and Ig2 isotypes specific for CL1 and CL2. Compared with unvaccinated goats, all three mimotopes dramatically reduced the number of parasites when the immunized goats were challenged with *F. hepatica* metacercariae. The mimotope CL1 reduced infection more effectively (by more than 70%) than the other two mimotopes. Additionally, phage immunization reduced fecal egg production, egg viability, and total parasite biomass [42]. In another study, the mimotopes TPWKDKQ of CL1 and YGSCFLR of CL2 induced high levels of IgG antibodies against CL1 and CL2. Compared with the control group, the CL1 vaccine gave 57.58% protection. However, CL2 or a combination of CL1 and CL2 had no significant effect in immunized animals (33.14% and 11.63%, respectively). Animals receiving CL2 have much lower parasite egg production [127]. The mimotopes SGTFLFS and WHVPRTWWVLPP of CL1 and the immunodominant E/S product displayed on phages induced the production of IgG1, IgG2, and IFN but reduced the levels of IL-4. The results indicated that the amino acids in the center and C-terminal end of the linear sequence are involved in the protection of sheep challenged with metacercariae of *F. hepatic* [98].

#### 3.9.3. *Plasmodium falciparum*

Malaria is a mosquito-borne illness caused by *Plasmodium falciparum*, *Plasmodium vivax*, *Plasmodium malariae*, *Plasmodium ovale*, or *Plasmodium knowlesi*. More than 90% of malaria deaths worldwide are caused by *P. falciparum*. Therefore, this parasite represents a public health concern worldwide [128]. Pfs47 is a 6-cysteine protein associated with the surface of Plasmodium gametes and mediates parasite evasion of the mosquito immune system, therefore, a potential vaccine target [129]. A phage-based vaccine called VLP-P47 was designed by conjugating a fifty-eight amino acid peptide from domain two of Pfs47 to AP205 VLP using the SpyCatcher. This study demonstrated that the VLP-P47 vaccine was more immunogenic than the unconjugated P47. BALB/c mice immunized intramuscularly with VLP-P47 resulted in considerably higher antibody titers than natural Pfs47 or unconjugated P47. These antibodies have high (83–98%) transmission-reducing activity (TRA), which indicates a level of reduction in oocysts compared to controls [51].

### 3.10. Phage-Based Vaccines against Yeast Infections

*Candida albicans* lives in small amounts on the skin, mouth, throat, gut, and vagina without causing problems. However, sometimes *C. albicans* becomes an opportunistic pathogen that threatens human health, particularly in immunocompromised individuals [130]. Fructose-bisphosphate aldolase 1 (Fba1) is a cell wall protein that is a potential anti-fungal target because yeast needs it for its growth [131]. Previously, it was shown that the peptide from Fba1 protected from candidiasis [132,133]. Phage-based vaccine against *C. albicans* was developed by displaying the Fba1 epitope YGKDVKDLFDYAQE on the coat proteins (pIII or pVIII) of filamentous phage (phage-3F and phage-8F). Immunized mice with phage-3F and phage-8F generated humoral and cellular immune responses. The vaccine reduces fungal infection, relieves kidney damage in infected mice, and notably improves their survival rates [99].

### 3.11. Phage-Based Vaccines against Cancer

#### 3.11.1. Breast Cancer

The human epidermal growth factor receptor-2 (HER2) is a biomarker of breast cancer that is often related to the poor prognosis of this disease [134]. However, anti-HER2 monoclonal antibodies help increase the overall survival of patients with HER2-positive breast cancer [135]. Therefore, HER2 is a vaccine target against this disease. A phage-based vaccine platform was developed by displaying HER2 and its variant Δ16HER2 on M13 phages, fused with pIII. Δ16HER2 transgenic mice received intraperitoneally either HER2-phage or Δ16HER2-phage. Moreover, vaccination with anti-HER2 phage-based vaccines could control breast cancer growth in ∆16HER2 mice. Anti-HER2 antibodies from immune sera have anticancer effects by impairing extracellular signal-regulated kinase (ERK) phosphorylation in human breast cancer cells (BT-474) [43]. A highly immunogenic peptide (GP2) derived from the transmembrane domain (654–662: IISAVVGIL) of HER2/neu protein was expressed in fusion with the capsid protein (gpD) in bacteriophage lambda. BALB/c mice were immunized with phages displaying GP2. The vaccine induced significant secretion of IFN-γ in the mice and cytotoxic T lymphocyte activity in a mouse tumor model over-expressing HER2/neu. It has a prophylactic effect by significantly reducing the growth rate of the tumor and prolonging survival compared with the control group. The GP2 phage-based vaccine has a therapeutic effect in a BALB/c mouse xenograft tumor model by efficiently inhibiting tumor growth and prolonging survival [100]. In another study, phage lambda displaying the E75 peptide from HER2, named λF7, was used to immunize mice in an implantable TUBO breast tumor model. Vaccinated and unvaccinated animals developed tumors in in vivo prophylactic experiments. In the therapeutic analysis, however, there was a substantial difference in tumor growth between mice inoculated with λF7 and control groups. Furthermore, mice inoculated with λF7 had longer survival periods [35].

xCT is a six extracellular domain protein found in different types of tumors, which can elicit anti-tumor antibodies in mouse models of breast cancer [136]. Phage-based vaccine against metastatic breast cancer (MBC) was developed by displaying an extracellular loop of xCT on VLPs of the bacteriophage MS2 (AX09). Mice immunized with AX09 produced high levels of specific antibodies against xCT. The sera from these animals reduced the number and size of the tumors, cystine uptake, and migration of breast cancer cells in vitro. In breast cancer models, AX09 immunization suppresses the generation of lung metastases and tumor progression [101].

#### 3.11.2. Other Types of Cancer

A personalized vaccination platform against cancer has been developed using phage display technology. Moreover, tumor-specific antigens presented on the surface of M13 phages can suppress tumors and their progression in tumor-bearing rats. In tumor-specific mutation models, the vaccine based on phages generated a strong neoantigen-based specific immune response. They prevented tumor recurrence after surgery in a therapeutically relevant surgical model while eliciting a long-term immunological memory effect. According to these results, the M13 is an effective tool for developing a bio-activated hybrid platform for individualized treatment [137].

The epidermal growth factor receptor (EGFR) is involved in different types of carcinoma (e.g., breast, lung, prostate, and head and neck cancer) and has been related to cancer progression and poor prognosis [138]. A previous study showed that vaccination with the extracellular domain of murine EGFR reduced lung metastasis in mice challenged with tumor cells [139]. The extracellular domain L2 of EGFR was expressed as a fusion peptide with pVIII protein on the surface of M13 phages. BALB/c mice immunized with the phage-based vaccine generated high titers of antibodies against EGFR L2 compared with the control groups. The prophylactic and therapeutic anticancer effects of the EGFR L2 vaccine were evaluated in a mouse tumor model. The results showed that this vaccine could reduce the tumor growth rate in immunized mice. Because the study was conducted on a few animals, it should be expanded to a larger number to obtain more solid and definitive results [140].

Disialoganglioside GD2 is a tumor-associated antigen that is an excellent target for cancer treatment [141]. GD2 has been studied as a candidate for an anticancer vaccine, but it is weakly immunogenic [142]. Additionally, acetylated GD2 may be used in vaccines due the hydroxyl groups of gangliosides may be acetylated in nature [143]. Anticancer phage-based vaccine was developed by conjugating a GD2 derivative bearing an N-acetamide at the C-9 position (9NHAc-GD2) to bacteriophage Qβ. After being inoculated with the Q-9NHAc-GD2 vaccine, mice developed robust and long-lasting IgG antibodies that were highly specific for 9NHAc-GD2 with limited cross-recognition of GD2. Canine immunotherapy with Q-9NHAc-GD2 revealed that the phage vaccine was immunogenic in dogs and not toxic [102].

Human mucin-1 (MUC1) is expressed all around the surface of cells. However, it goes under glycosylation by most human tumor cells [144]. The extracellular domain of MUC1 contains a variable region of twenty amino acid tandem repeats (HGVSTAPDTRPAPGSTAPPA) with five potential O-glycosylation sites that become exposed in cancer cells [145]. Clinical investigations show that the presence of MUC1 antibodies is related to a better prognosis of the disease. However, there is currently no effective MUC1 vaccine [146]. Phage-based vaccine for anticancer immunotherapy was developed by covalently linking the glycosylated peptide HGVSTAPDTRPAPGSTAPPA to the bacteriophage Qβ, named Qβ-MUC1. Immunized MUC1 transgenic mice with the Qβ-MUC1 vaccine produced high levels of antibodies against MUC1, exhibiting tumor protection in metastasis and solid tumor models. Antibodies from sera from Qβ-MUC1 immunized mice recognized tissues from human breast cancer but not from normal human breasts [103].

Neoantigens are foreign antigens that boost the immune response against cancer cells by effectively activating the T-cell response [147]. Phage-based vaccine against cancer cells were developed by expressing neoepitopes from mutated proteins of B16-F10 melanoma tumor cells. In mice, a dose of phage-peptide vaccines elicits a high immune response. Induced antibodies bound to the mutant peptides more strongly than the equivalent not mutated peptides. In comparison with uninfected mouse nodes, next-generation sequencing (NGS) of lymph node cells revealed poor B-cell receptor diversity and clonal hyperpolarization in vaccine-draining lymph nodes. Following immunization, the NGS data indicated a rise in IgG and other class-switched antibodies. These findings agree with previous data on plasma IgG antibodies against peptides on T7 phages that recognize whole B16-F10 cells [37].

### 3.12. Phage-Based Vaccines against Other Types of Diseases or Conditions

Tauopathies are a group of neurodegenerative diseases, including Alzheimer’s disease (AD) and frontotemporal dementia (FTD). Hyperphosphorylated pathological tau (pTau) forms neurofibrillary tangles (NFTs) in the neurons [148], causing Tauopathies. Therefore, Tau may be a good target in a vaccine against these diseases. A phage-based vaccine against Tauopathies (pT181-Qβ) was developed by conjugating a peptide from Tau (175TPPAPKpTPPSSGEGGC190; named pT181) to VLPs of bacteriophage Qβ. Tauopathy-prone transgenic rTg4510 and non-transgenic mice vaccinated with pT181-Qβ generated a robust IgG response against pT181. These antibodies could detect classical somatodendritic pTau+ in the brains of FTD but not in the brain sections of healthy controls. The pTau accumulated in the hippocampus and cortex was reduced in pT181-Qβ vaccinated mice. Neuroinflammation and circulating CD3+ T-cells in the brain were also reduced [104].

Hypertension is a heart disease risk factor [149], and the human L-type calcium channel (CaV1.2) is the most prominent voltage-gated channel type that regulates Ca^2+^ influx in vascular smooth muscle. Changes in CaV1.2’s molecular structure and function generate vascular dysfunction and hypertension [150]. A phage-based vaccine for treating hypertension was created by coupling the epitope of the E3 region domain IV (CE12) of CaV1.2 to VLPs of the bacteriophage Qβ. The Q-CE12 vaccination successfully elicited anti-CE12 antibodies, reducing blood pressure in hypertensive animals. Moreover, these antibodies recognize CaV1.2, blocking its function. Additionally, there was no evidence of adverse effects [46].

Allergic rhinitis (AR) affects 500 million people worldwide and is associated with severe morbidity, productivity loss, and healthcare expenses [151]. Also, four subtypes of histamine receptors (e.g., H1, H2, H3, and H4) have been reported so far. The blockage of the histamine H4 receptor (H4R) reduces AR symptoms and inflammation in animal models of AR [85]. A phage-based vaccine against AR was developed by displaying H4R epitopes on phage M13. This phage clone (P-FN12; FNKWMDCLSVTH) was selected from a 12-mer random peptide library using an affinity-purified anti-H4R antibody. Immunized rats with the P-FN12 vaccine induced the production of anti-H4R antibodies. The IgE and IL-4 levels decreased but increased the interferon (IFN)-γ and interleukin-2 (IL-2) levels in immunized rats. Additionally, the vaccine lowers the proportion of Th1/Th2 cells in PBMCs and inhibits the infiltration of eosinophils in the nasal mucosa. P-FN12 is a promising vaccine for treating AR [105].

Cardiovascular diseases (CVDs) are a collection of heart and blood vessel ailments that kill over seventeen million people worldwide [152]. Low-density lipoprotein cholesterol or LDL-C is associated with an increased risk of CVDs [153] and comprises three proteins (e.g., PCSK9, ApoB, and CETP) that are checkpoint proteins. A phage-based vaccine was developed by displaying these three proteins on the coat protein VLPs of bacteriophage Qβ. The phage-based vaccines were prepared in hot melt-extrusion to ensure a slow-release. Immunized mice generated immunoglobulin isotypes IgG1 and IgG2b against PCSK9, ApoB, and CETP. Additionally, the vaccines reduce the levels of these proteins, thus, decreasing the total cholesterol in the plasma [66].

The Gonadotropin-releasing hormones (GnRH) are peptide hormones produced in the hypothalamus, which are released to stimulate the anterior pituitary and release gonadotropin follicle-stimulating hormone (FSH) and luteinizing hormone (LH), which control ovarian and testicular function [154]. GnRH has been used in vaccine formulations, which limit reproduction and decrease sex hormone synthesis in various mammalian species, including cats, over lengthy periods [155]. A phage-based vaccine to control fertility in cats was selected from an f8-8 landscape phage display library with neutralizing GnRH antibodies [44]. The cats vaccinated with phage-GnRH produced high levels of antibodies against GnRH, decreasing their serum testosterone levels and reducing their total testicular volume, indicating gonadal atrophy. Moreover, all cats generated sperm at the end of the trial. However, their normal morphology was reduced by up to 38%. These results demonstrate that the phage-based vaccination against GnRH may be used to suppress cat fertility [45].

## 4. Discussion

In this review, we have answered several questions about phage-based vaccines by conducting a literature search. Vaccine development is a long and complicated process. An important issue is that an effective delivery method and an adjuvant are necessary for vaccines to produce strong immune reactions. Therefore, a phage-based vaccine platform might be an ideal tool for vaccine development because phages have several important features. First, phages are not infection agents for eukaryotes. Therefore, they cannot induce diseases in humans or animals. Second, phages show adjuvant properties. Accordingly, adjuvants are not needed. Moreover, the side effects of adjuvants are avoided. Third, the phage genome can be easily manipulated to create a fusion of antigens to the phage coat proteins or to create phage DNA vaccines. In addition, phages can be decorated with proteins or antigen subunits. Fourth, phages are stable under different conditions, which can facilitate phage-based vaccine transport and storage that will be beneficial for vaccine delivery to remote locations or areas with limited resources. Fifth, phages are easy to produce on a large scale, which might reduce production costs and can be affordable to developing and low-income countries. Furthermore, a phage-based vaccine system could be implemented in these countries so that they can produce vaccines according to their needs. Sixth, phage-based vaccines have applications for several diseases or conditions (e.g., infectious diseases, cancer, neurodegenerative diseases, cardiovascular diseases, allergies, and fertility control). Other issues, such as administration route and phage clearance from the body, have not been mentioned here because reviews have already addressed them [156,157]. Despite the excellent phage vaccine characteristics and substantial preclinical data, the FDA and EMA have not yet approved any bacteriophage vaccines.

In addition, phages are viewed skeptically in the U.S. and other countries while phages in therapy have been used for decades in eastern European countries. Probably, one of the main reasons for this is that we do not know the secondary effects of bacteriophages in the long-term and there is no clinical study that has done a long-term follow-up. The reason is that bacteriophages contain several components such as phage genome and capsid proteins that make phages uncertain as a part of a vaccine since we lack information about whether such components are deposited or affect the human or animal body. However, most of the vaccines used so far have been multi-component vaccines based on pathogens in live and attenuated vaccines or VLPs as delivery vehicles. VLPs are particles composed of capsid proteins that do not contain genetic material but contain capsid proteins. The FDA has approved several VLP-based vaccines against infectious diseases, whereas VLPs based on bacteriophages are under clinical trials [26]. Another factor that probably meets resistance to the acceptance of phages in vaccines is that phages can kill bacteria and could kill bacteria from the human flora. However, there are data from a human trial showing that phages do not affect the human flora and that oral administration of phages is safe [58]. In addition, the natural microbial community of the human gut is frequently exposed to bacteriophages in high concentrations through food, water, and the environment without causing any harm to people [68]. Another issue that can create concern and probably skepticism towards approving phage-based vaccines is endotoxin contamination during phage production and its secondary effects. However, there are effective protocols to remove endotoxins like the one used by Roehnisch et al., 2014 [13]. They demonstrated that their M13-based vaccine was well-tolerated by fifteen patients with advanced multiple myeloma, with only modest and temporary side effects, such as flu-like symptoms and skin irritability at the injection site. The use of phages for medical purposes, such as phage therapy, was long resisted by the Western world, but this has started to change. New technologies, therefore, undergo a protracted and slow process before being accepted and authorized if they meet the standards for safety and efficacy, especially those designed for clinical application.

Surprisingly, many encouraging preclinical studies that could have advanced to clinical trials have not been repeated, and we do not know why. Therefore, additional efforts are needed to bring this technology with great potential and many benefits to the next level of clinical trials to provide proof of efficacy for their use in humans and animals. The perspective of phage-based vaccines is good because phages are safe for humans and animals, which has been proven in several clinical trials. Bacteriophages are approved as antibacterial additives in food products and for emergency use in phage therapy to treat bacterial infectious diseases, but not in vaccines. The promising results from the clinical trial I/II of Roehnisch et al., 2014 [13], should be a good starting point to increase phage-based vaccines in clinical trials, at least those based on M13 that have shown to be safe. This perspective is encouraging because the above may indicate that the application of phages in vaccines could be authorized in the future.

## Figures and Tables

**Figure 1 vaccines-11-00436-f001:**
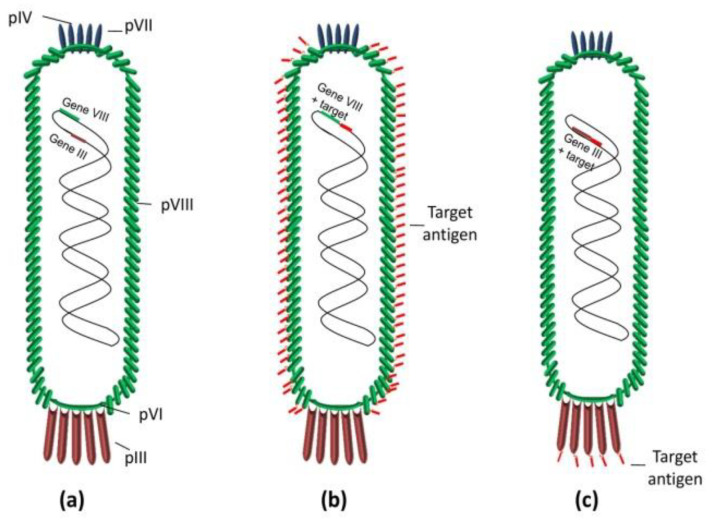
Scheme of bacteriophage M13: (**a**) wild-type phage; (**b**) phage displaying multivalent copies of the target antigen on pVIII protein; and (**c**) phage displaying multivalent copies of the target antigen on phage pIII protein. In both cases, the DNA fragments encoding the target antigens were cloned into a phage vector.

**Figure 2 vaccines-11-00436-f002:**
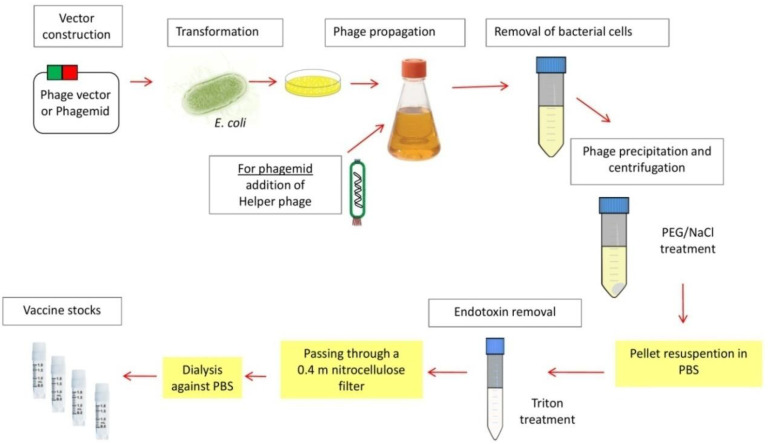
Schematic of the preparation for M13 phage-based vaccine.

## Data Availability

The data is available in the articles included in this review.

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
