# Peer review of "Aspects of Phage-Based Vaccines for Protein and Epitope Immunization"

_vaccines, 2023, doi:10.3390/vaccines11020436_

Round 1

Reviewer 1 Report

In the manuscript "Aspects of phage-based vaccines for protein and epitope immunization", Palma systemically reviewed the advances of phage-based vaccines for pontential applications. The paper is well written, and gives the readers comprehensive information about phage-based vaccines. The manuscript could be considered for acceptance after minor revision:

(1) Besides the phages of E. coli and Acinetobacter, are there other kinds of phages that could be used for vaccine development? This should be further introduced.

(2) Whether do the phages carry the components of the host bacteria that may evoke non-specific immunity reaction? This should be discussed.

(3) One or two schematic images should be added to illustrate construction and purification of vaccine-aimed phages, which may be easily understood.

(4) Why are the phage-based vaccines hard to be in real application? This should be further discussed in the paper.

(5) Do the phages directly used as vaccines? or other adjuvants are needed for prepartion of the final vaccines? A common preparation of phage-based vaccines should be introduced.

Author Response

Dear reviewer,

Thanks for reviewing my manuscript.

I have answered your questions below and made the suggested changes.

Comments and Suggestions for Authors

In the manuscript "Aspects of phage-based vaccines for protein and epitope immunization", Palma systemically reviewed the advances of phage-based vaccines for potential applications. The paper is well-written, and gives the readers comprehensive information about phage-based vaccines. The manuscript could be considered for acceptance after minor revision:

(1) Besides the phages of E. coli and Acinetobacter, are there other kinds of phages that could be used for vaccine development? This should be further introduced.

Answer: Thanks for the interesting question. I clarified this issue in the manuscript lines 76-92.

“For the production of vaccines, several lytic and filamentous phages have been used, especially those that replicate in E. coli [11]. In theory, any bacteriophage is suitable for phage-based vaccines, but they should meet some criteria to facilitate phage construction and production. The sequence of the phage genome and the genes encoding the capsid and tail proteins should be known and can be modified to promote the display of foreign antigens [12]. The incorporation of the target sequence should not affect the phage stability and assembly. The phage bacterial host should be a nonpathogenic laboratory strain that can be used on a standard laboratory bench. Also, an alternative approach could be used, which applies to any bacteriophage, and antigens can be covalently linked to the bacteriophage surface by a chemical method such as the SpyCatcher system. However, the toxicity of untested bacteriophages should be analyzed first. One of the main reasons that E. coli phages are used for phage-based vaccines is that they are well-known, easy to use, and have already been tested in animal models and clinical trials [13]. In addition, many well-developed tools and protocols are available for engineering these phages. The most common phages used in the selected studies were lambda, Qb, T4, MS2, T7, AP205, M13, and M13KE, which are listed in table 1.”

(2) Whether do the phages carry the components of the host bacteria that may evoke non-specific immunity reaction? This should be discussed.

Answer: Thanks for taking up such an important issue for vaccine development. To address this, I added some points in lines 164 -170, 303-320, and 751-757.

Lines: 164 -170

“To employ phage preparations in vaccines, they must be cleaned of bacterial endotoxins, which are frequently present when it uses bacterial cultures. According to Bordier (1980), contaminating endotoxins are eliminated by repeated two-phase Triton X-114 separation [53]. The cleaned bacteriophages are dialyzed against sterile PBS after passing through a 0.4 m nitrocellulose filter. Other endotoxins removal methods could be used such as the one with organic solvents described by Szermer-Olearnik and BoratyÅ„ski (2015)[54].”

Lines: 303-320

“Endotoxins (lipid A), released during the replication of lytic phages [59] or derived from the host bacteria and culture media, can contaminate the production of phage-based vaccines. Endotoxins are also the hydrophobic moiety of lipopolysaccharide (LPS), which makes up the outer monolayer of the outer membranes of most gram-negative bacteria [88]. Endotoxins can be an issue since even little exposure can trigger pro-inflammatory reactions [88] and lead to toxic shock, cell damage, and the production of cytokines [89]. Due to these effects, the generated phage stock must undergo a procedure to remove endotoxins before they can be used in vaccines. This can be minimized using lysogenic phages such as filamentous phages (e.g., M13, fd, and f1) stead of lytic phages and implementing Good Manufacturing Practice (GMP)[90] in an early stage of the product development to meet regulatory requirements [91]. The removal of endotoxins from bacterial cultures can be done using many techniques, such as triton X-114 separation [53] and organic solvents [54]. The triton X-114 separation worked quickly and effectively to prepare M13 phage-based vaccine by lowering the endotoxin concentration to less than one unit/ml, which was used in patients with advanced multiple myeloma participating in a clinical phase I/II trial. The triton X-114 treated phage vaccination was well-tolerated in the patients, with the only minor side effects being flu-like symptoms and skin irritation at the injection site [13].”

Lines: 752-758

“Another issue that can create concern and probably skepticism toward approving phage-based vaccines is endotoxin contamination during phage production and its secondary effects. However, there are effective protocols to remove endotoxins like the one used by Roehnisch et al., 2014. They demonstrated that their M13-based vaccine was well-tolerated by fifteen patients with advanced multiple myeloma, with only modest and temporary side effects, such as flu-like symptoms and skin irritability at the injection site.”

 (3) One or two schematic images should be added to illustrate construction and purification of vaccine-aimed phages, which may be easily understood.

Answer: Thanks for the suggestions

Two pictures were added to the manuscript; one shows the two most common types of display in bacteriophage M13 (fig.1), and the other shows the main steps of the M13 phage-based vaccine production (fig. 2).

The production of M13 is explained at the end of section 3.1, lines 128-177.

“Phage-based vaccine preparation entails several steps, including the creation of plasmids, the transformation of E. coli, phage propagation, removal of bacterial cells, phage precipitation and centrifugation, pellet resuspension, endotoxin removal, filtration, dialysis, and aliquotation of the vaccine stock (shown in Fig. 2).

The propagation process for each phage and bacterial host may be different than another. Here is a brief description of the production of the most commonly used phage in phage-based vaccine development, the lysogenic filamentous phage M13, which can infect trains of F+ male E. coli through their pili (e.g., TG- 1). To display the target antigen on the surface of filamentous phages, a phage vector (e.g., fUSE5, fAFF1, fd-CAT1 or fd-tet-DOG) or a phagemid [52] (e.g., pHEN1, pComb3, pComb8, or pSEX) are used. A phage vector is composed of almost complete phage genome (e.g., M13), into which DNA encoding the desired antigen is introduced in the gene encoding a capsid protein (e.g., pVIII or pIII). The remaining portion of the phage genome typically remains untouched and generates the additional gene products required for the phage life cycle. The phage vectors allow multivalent expression of the target antigen, also displayed in fusion with every copy of the chosen capsid protein. A phagemid, on the other hand, is a plasmid that carries only the gene for one of the phage capsid proteins; therefore, a helper phage (e.g., M13KO7 and VCSM13 ) must provide the remaining phage proteins for phage production.

After being constructed, the phage vector or the phagemid is introduced into TG-1 by electroporation or chemical transformation.

For phage propagation from bacterial cells with phagemid, an overnight culture is used to inoculate a sterile growth medium in an Erlenmyer flask and grow at 37 °C to mid-log phase, at which point helper phage is used to infect the bacterial cells in the ratio of 1:20 and incubate in falcon tube without shaking in 37 °C water bath for 30 min. The infected cells are collected by centrifugation, then the supernatant is removed, the pellet resuspended in a fresh medium containing antibiotic, and incubated at 30 °C overnight on a shaker in a baffled flask. For propagating phages from bacterial cells with phage vectors, the step with helper phages is omitted.

Bacterial cells are removed from the culture containing the recombinant phages by centrifugation twice and discarding the pellets. Then, the phages are collected by treatment of the supernatant with 1/5 volume of 20% PEG/2.5 M NaCl for 2 h at 4 °C and centrifugation. The supernatant is then discarded and the pellet is reconstituted in sterile phosphate-buffered saline (PBS), and the PEG/NaCl procedure is carried out once more.

To employ phage preparations in vaccines, they must be cleaned of bacterial endotoxins, which are frequently present when it uses bacterial cultures. According to Bordier (1980), contaminating endotoxins are eliminated by repeated two-phase Triton X-114 separation [53]. The cleaned bacteriophages are dialyzed against sterile PBS after passing through a 0.4 m nitrocellulose filter. Other endotoxins removal methods could be used such as the one with organic solvents described by Szermer-Olearnik and Boratyński (2015)[54].

The quality of the stock needs to be checked before using it by determining its titer and endotoxin levels. Then, it can be diluted to the working concentration.

The prepared phage stocks can then be aliquoted into cryotubes as working stocks and temporarily stored at 4 °C. For long-term storage, glycerole is added to a final concentration of 20%, which can be kept at -80°C.

Before a frozen phage stock is used, the glycerol needs to be removed by treating it with PEG/NaCl and centrifugation and then resuspending the pellet in sterile PBS.”

(4) Why are the phage-based vaccines hard to be in real application? This should be further discussed in the paper.

Answer: Thanks for the suggestion

I addressed this in the discussion in lines 734-757. I added the issue of endotoxin contamination as another potential reason in lines 751-757.

Lines 735-758

“Also, phages are viewed skeptically in the U.S. and other countries while phages in therapy have been used for decades in eastern European countries. Probably, one of the main reasons is that we do not know the secondary effects of bacteriophages in the long- term and there is no clinical study that has done a long-term follow-up. The reason is that bacteriophages contain several components such as phage genome and capsid proteins that make phages uncertain as a part of a vaccine since we lack information about whether such components are deposited or affect the human or animal body. But, most of the vaccines used so far have been multi-component vaccines based on pathogens in live and attenuated vaccines or VLPs as delivery vehicles. VLPs are particles composed of capsid proteins that do not contain genetic material but contain capsid proteins. The FDA has approved several VLP-based vaccines against infectious diseases, whereaswhile VLPs based on bacteriophages are under clinical trials [26]. Another factor that probably meets resistance to the acceptance of phages in vaccines is that phages can kill bacteria and could kill bacteria from the human flora. However, there are data from a human trial showing that phages do not affect the human flora and that oral administration of phages is safe [58]. In addition, the natural microbial community of the human gut is frequently exposed to bacteriophages in high concentrations through food, water, and the environment without causing any harm to people [68]. Another issue that can create concern and probably skepticism toward approving phage-based vaccines is endotoxin contamination during phage production and its secondary effects. However, there are effective protocols to remove endotoxins like the one used by Roehnisch et al., 2014. They demonstrated that their M13-based vaccine was well-tolerated by fifteen patients with advanced multiple myeloma, with only modest and temporary side effects, such as flu-like symptoms and skin irritability at the injection site.”

(5) Do the phages directly used as vaccines? or other adjuvants are needed for prepartion of the final vaccines? A common preparation of phage-based vaccines should be introduced.

Answer: Thanks for taking this important issue.

I addressed this issue in lines 211-234

“Phages-based vaccines commonly do not require adjuvants in their composition because bacteriophages are effective adjuvants, capable of increasing the immunological response to any antigen administered together with the phage particles. Unmethylated deoxycytidylate-phosphate-deoxyguanylate (CpG) dinucleotide in the phage genome induces this adjuvant effect [63]. Also, the unmethylated CpG nucleotides can induce murine B-cells to proliferate and secrete immunoglobulin [64], induce cytokines, stimulate natural killer cells, and elicit T-cell responses [65].

Recently, Davenport and colleagues demonstrated that the recombinant receptor-binding domain (RBD) of the severe acute respiratory syndrome (SARS) alone was insufficient to induce RBD-specific IgG in mice; however, RBDSARS- phage-like particles from lambda decorated with RBD not only elicited high levels of RBDSARS-specific IgG, but they remained high for 174 days [34]. In the study by Ortega-Rivera et al., mice immunized with Qβ conjugated with the S protein peptides TESNKKFLPFQQFGRDIA, PSKPSKRSFIEDLLFNKV, HADQLTPTWRVY produced high-level IgG antibodies, while animals vaccinated with free S protein peptides did not [47]. Li and colleagues found that whereas the proteins of influenza virus 3M2e displayed on T4 nanoparticles, without any adjuvant, elicited extremely high- levels of 3M2e-specific IgG antibodies, whereas 3M2e conjugated to RB69 Soc induced low levels of 3M2e-specific IgG antibodies [38]. 58 amino acid peptide of the Pfs47 antigen of P. falciparum conjugated phage AP205 VLP was more immunogenic than unconjugated Pfs47 monomers, inducing a high level of antibodies [51]. VLPs of bacteriophage Qβ displaying the target peptides PCKS9, ApoB, and CETP induced higher antibody titers in mice against the antigen than the free peptides [66].

In some studies, an adjuvant was also used along with phages [42,67], perhaps because they needed a higher immune response than that obtained with only the phages.”

I mentioned this in the discussion, lines 718-719

“Second, phages show adjuvant properties; therefore, adjuvants are not needed. Also, the side effects of adjuvants are avoided.”

A common preparation of phage-based vaccines has been added in lines 128-177 and fig. 2.

Thanks!

Reviewer 2 Report

The review article entitled “Aspects of phage-based vaccines for protein and epitope immunization” by Marco Palma is a compilation of the application of phages as vaccine delivery vehicles. The author has covered several diseases and pathogens to highlight the ongoing research showing the safety and benefits of a phage based vaccine. The author highlights the advantages of using phages for vaccine development and also provides the hesitation of letting these vaccine move forward from the preclinical stage. Elaborating more on the disadvantages on these vaccines would be beneficial for the review. As these kind of vaccines are approved for emergency use what are the authors view on the plausible holdup allowing them to move on to the next stage. The review is well written but can benefit with some editing to improve the presentation of the article making it more interesting for the readers.    

Author Response

The review article entitled “Aspects of phage-based vaccines for protein and epitope immunization” by Marco Palma is a compilation of the application of phages as vaccine delivery vehicles. The author has covered several diseases and pathogens to highlight the ongoing research showing the safety and benefits of a phage based vaccine. The author highlights the advantages of using phages for vaccine development and also provides the hesitation of letting these vaccine move forward from the preclinical stage. Elaborating more on the disadvantages on these vaccines would be beneficial for the review. As these kind of vaccines are approved for emergency use what are the authors view on the plausible holdup allowing them to move on to the next stage. The review is well written but can benefit with some editing to improve the presentation of the article making it more interesting for the readers.

Thanks for the interesting points:

Answers:

  • I addressed the disadvantages in lines 292-302 and in addition added the issue of endotoxin in phage preparation in lines 164-167, and in discussion in lines 303-320

lines 292-302

“Phage-based vaccines have some potential limitations that need to be mentioned and considered when developing them. For example, it can be challenging to correctly display a molecule on the phage surface, resulting in missing active epitopes to elicit a meaningful immune response [84]. In addition, the antigen size can be a limitation because it is unlikely that large protein subunits could be displayed on phage particles; however, this is solved by conjugating the phages with these large antigens. Additionally, for the phage DNA vaccines, the genome length must be within the virion packaging limits [86]. The lack of an immunological response to self or harmful antigens may also be a barrier [87]. Another disadvantage is that few clinical trials support the efficacy of phage-based vaccines, and more efforts must convert the results from preclinical investigations to clinical trials.”

Lines 164-170

“To employ phage preparations in vaccines, they must be cleaned of bacterial endotoxins, which are frequently present when it uses bacterial cultures. According to Bordier (1980), contaminating endotoxins are eliminated by repeated two-phase Triton X-114 separation [53]. The cleaned bacteriophages are dialyzed against sterile PBS after passing through a 0.4 m nitrocellulose filter. Other endotoxins removal methods could be used such as the one with organic solvents described by Szermer-Olearnik and BoratyÅ„ski (2015)[54].”

Lines 303-320

“Endotoxins (lipid A), released during the replication of lytic phages [59] or derived from the host bacteria and culture media, can contaminate the production of phage-based vaccines. Endotoxins are also the hydrophobic moiety of lipopolysaccharide (LPS), which makes up the outer monolayer of the outer membranes of most gram-negative bacteria [88]. Endotoxins can be an issue since even little exposure can trigger pro-inflammatory reactions [88] and lead to toxic shock, cell damage, and the production of cytokines [89]. Due to these effects, the generated phage stock must undergo a procedure to remove endotoxins before they can be used in vaccines. This can be minimized using lysogenic phages such as filamentous phages (e.g., M13, fd, and f1) stead of lytic phages and implementing Good Manufacturing Practice (GMP)[90] in an early stage of the product development to meet regulatory requirements [91]. The removal of endotoxins from bacterial cultures can be done using many techniques, such as triton X-114 separation [53] and organic solvents [54]. The triton X-114 separation worked quickly and effectively to prepare M13 phage-based vaccine by lowering the endotoxin concentration to less than one unit/ml, which was used in patients with advanced multiple myeloma participating in a clinical phase I/II trial. The triton X-114 treated phage vaccination was well-tolerated in the patients, with the only minor side effects being flu-like symptoms and skin irritation at the injection site [13].”

  • I clarified something about the emergency use of phages in the discussion, lines 768-773.

“Bacteriophages are approved as antibacterial additives in food products and for emergency use in phage therapy to treat bacterial infectious diseases, but not in vaccines. The promising results from the clinical trial I/II of Roehnisch et al., 2014, should be a good starting point to increase phage-based vaccines in clinical trials, at least those based on M13 that have shown to be safe.”

  • I added two new figures to the manuscript: one showing the structure of phage M13 and the display of target antigen using phage capsid protein pVIII and pIII, and a second figure showing the steps for based-vaccine preparation focus on phage M13. In addition, text explaining the phage preparation for vaccines was added.

Thanks!